# Characterization of TSET, an ancient and widespread membrane trafficking complex

Jennifer Hirst[1]*[†], Alexander Schlacht[2†], John P Norcott[3‡], David Traynor[4‡], Gareth Bloomfield[4], Robin Antrobus[1], Robert R Kay[4], Joel B Dacks[2]*, Margaret S Robinson[1]*

[1]Cambridge Institute for Medical Research, University of Cambridge, Cambridge, United Kingdom; [2]Department of Cell Biology, University of Alberta, Edmonton, Canada; [3]Department of Engineering, University of Cambridge, Cambridge, United Kingdom; [4]Cell Biology, MRC Laboratory of Molecular Biology, Cambridge, United Kingdom

**Abstract** The heterotetrameric AP and F-COPI complexes help to define the cellular map of modern eukaryotes. To search for related machinery, we developed a structure-based bioinformatics tool, and identified the core subunits of TSET, a 'missing link' between the APs and COPI. Studies in *Dictyostelium* indicate that TSET is a heterohexamer, with two associated scaffolding proteins. TSET is non-essential in *Dictyostelium*, but may act in plasma membrane turnover, and is essentially identical to the recently described TPLATE complex, TPC. However, whereas TPC was reported to be plant-specific, we can identify a full or partial complex in every eukaryotic supergroup. An evolutionary path can be deduced from the earliest origins of the heterotetramer/scaffold coat to its multiple manifestations in modern organisms, including the mammalian muniscins, descendants of the TSET medium subunits. Thus, we have uncovered the machinery for an ancient and widespread pathway, which provides new insights into early eukaryotic evolution.

*For correspondence: jh228@ cam.ac.uk (JH); dacks@ualberta.ca (JBD); msr12@cam.ac.uk (MSR)

[†]These authors contributed equally to this work

[‡]These authors also contributed equally to this work

Competing interests: The authors declare that no competing interests exist.

## Introduction

The evolution of eukaryotes some 2 billion years ago radically changed the biosphere, giving rise to nearly all visible life on Earth. Key to this transition was the ability to generate intracellular membrane compartments and the trafficking pathways that interconnect them, mediated in part by the heterotetrameric adaptor complexes, APs 1–5 and COPI (*Dacks et al., 2008*; *Field and Dacks, 2009*; *Hirst et al., 2011*; *Koumandou et al., 2013*). In mammals, APs 1 and 2 and COPI are essential for viability, while mutations in the other APs cause severe genetic disorders (*Boehm and Bonifacino, 2002*; *Hirst et al., 2013*). The AP and COPI complexes share a similar architecture, due to common ancestry predating the last eukaryotic common ancestor (LECA). All six complexes consist of two large subunits of ~100 kD, a medium subunit of ~50 kD, and a small subunit of ~20 kD (*Figure 1A*). Their function is to select cargo for packaging into transport vesicles, and together with membrane-deforming scaffolding proteins such as clathrin and the COPI B-subcomplex, they facilitate the trafficking of proteins and lipids between membrane compartments in the secretory and endocytic pathways. The recent discovery of the evolutionarily ancient AP-5 complex, found on late endosomes and lysosomes, added a new dimension to models of the endomembrane system, and raised the possibility that other undetected membrane-trafficking complexes might exist (*Hirst et al., 2011*). Therefore, we set out in search of additional members of the AP/COPI subunit families.

**eLife digest** Eukaryotes make up almost all of the life on Earth that we can see around us, and include organisms as diverse as animals, fungi, plants, slime moulds, and seaweeds. The defining feature of eukaryotes is that, unlike nearly all bacteria, they have membrane-bound compartments—such as the nucleus—within their cells.

Moving molecules, such as proteins, between these compartments is essential for living eukaryotic cells, and these molecules are usually trafficked inside membrane-bound packages called vesicles. Two similar sets of protein complexes—each containing four different subunits—ensure that the molecules are packaged inside the correct vesicles. However, it is not clear how these two protein complexes (called the AP complexes and the COPI complex) are related to each other, and when and where they originated in the history of life.

Now, Hirst, Schlacht et al. have discovered a new—but very ancient–protein complex that they refer to as the 'missing link' between the AP and COPI complexes. The four subunits inside this new complex were found by searching for proteins with shapes that were similar to those of the AP and COPI proteins, rather than just searching for proteins with similar sequences of amino acids. This approach identified related protein subunits in groups as diverse as plants and slime moulds, which suggests that this protein complex evolved in the earliest of the eukaryotes. The four subunits identified in a slime mould were confirmed to interact, and also shown to bind to the plasma membrane of living cells.

One of the subunits had already been named TPLATE, so Hirst, Schlacht et al. decided to call the complex TSET; the other three subunits were named TSAUCER, TCUP and TSPOON, and two other proteins that interacted with the complex were both called TTRAY.

While most of the TSET complex itself has been lost from humans and other animals, one of subunit appears to have evolved into a family of proteins that help molecules get into cells. The discovery of TSET reveals another major player in vesicle-trafficking that is not only important for our understanding of how modern eukaryotes work, but also how ancient eukaryotes evolved.

## Results and discussion

### The search for novel AP-related complexes

Because we were unable to find any promising candidates for new AP/COPI-related machinery using sequence-based searches, we developed a more sensitive tool, designed to search for structural similarity rather than sequence similarity. Using HHpred to analyse every protein in the RefSeq database from 15 organisms, covering a broad span of eukaryotic diversity, we built a 'reverse HHpred' database. This database contains potential homologues for >300,000 different proteins (http://reversehhpred.cimr.cam.ac.uk), and can be searched with structures from the Protein Data Bank (PDB). As proof of principle, we used this database to identify all four subunits of the AP-5 complex (*Figure 1—figure supplements 1 and 2*; *Figure 1—source data 1, 2*), even though in our previous study only the medium subunit was initially detectable by bioinformatics-based searching (*Hirst et al., 2011*).

In addition to known proteins, our reverse HHpred database revealed novel candidates for each of the four subunit families, with orthologues present in diverse eukaryotes including plants and *Dictyostelium* (*Figure 1—figure supplements 2–4*, *Figure 1—source data 1, 2*). Secondary structure predictions confirmed that the new family members have similar folds to their counterparts in the AP complexes and COPI (*Figure 1B*). Only one of these proteins had been characterised functionally: TPLATE (NP_186827.2), an *Arabidopsis* protein related to the AP β subunits and β-COP, found in a microscopy-based screen for proteins involved in mitosis and localised to the cell plate (*Van Damme et al., 2006*; *Van Damme et al., 2011*). There is some variability between orthologous subunits in different organisms: for instance, *Arabidopsis* has added an SH3 domain to the C-terminal end of its 'γαδεζ' large subunit, while *Dictyostelium* has lost the μ homology domain (MHD) at the end of its medium subunit; and in general there seems to be much less selective pressure on these genes than on those encoding other AP/COPI family members (e.g., the AP-1 β1 subunits are 58.01% identical in *Dictyostelium* and *Arabidopsis*, while the new β family members are only 14.63% identical).

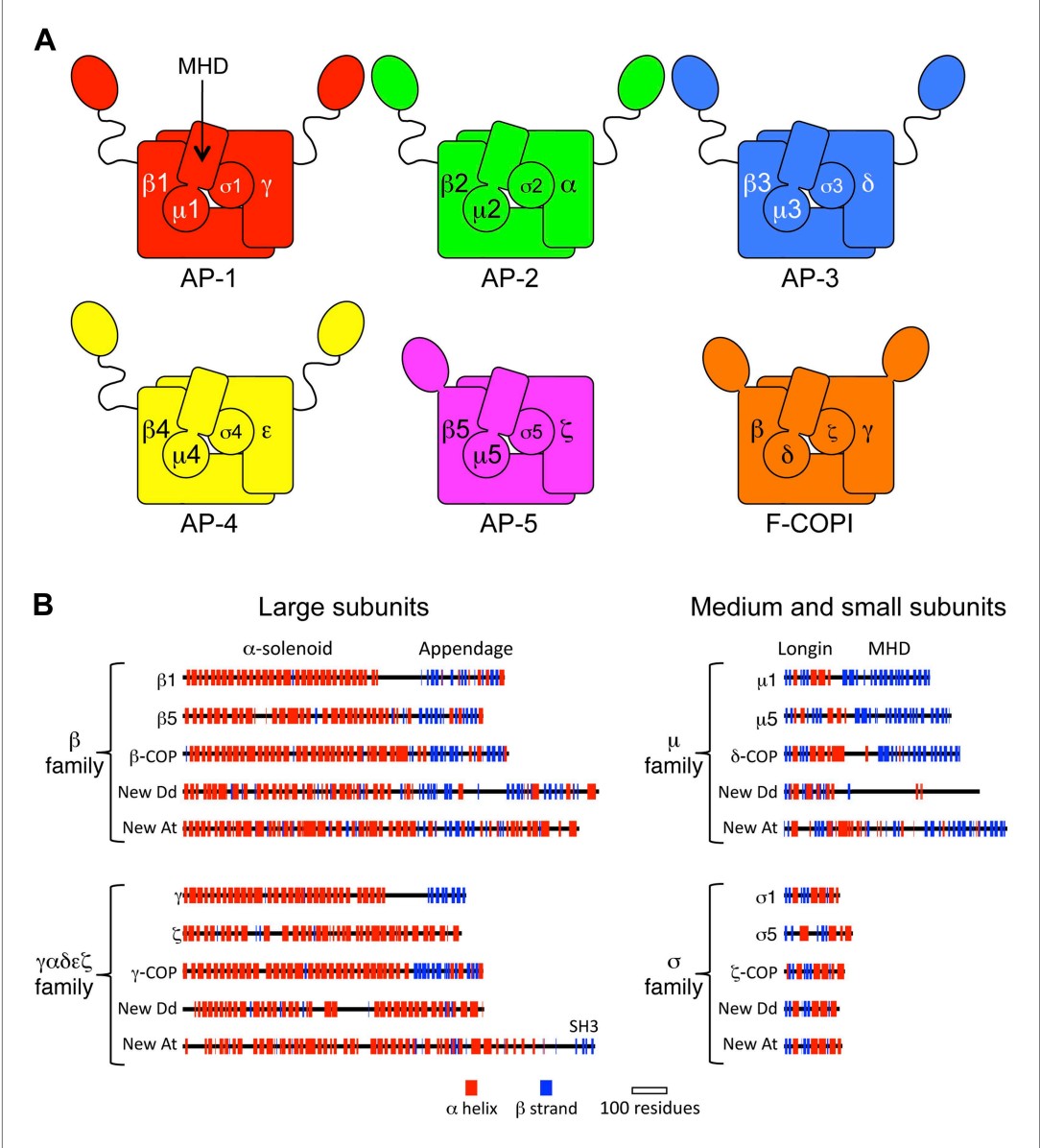

**Figure 1**. Diagrams of APs and F-COPI. (**A**) Structures of the assembled complexes. All six complexes are heterotetramers; the individual subunits are called adaptins in the APs (e.g., γ-adaptin) and COPs in COPI (e.g., γ-COP). The two large subunits in each complex are structurally similar to each other. They are arranged with their N-terminal domains in the core of the complex, and these domains are usually (but not always) followed by a flexible linker and an appendage domain. The medium subunits consist of an N-terminal longin-related domain followed by a C-terminal μ homology domain (MHD). The small subunits consist of a longin-related domain only. (**B**) Jpred secondary structure predictions of some of the known subunits (all from *Homo sapiens*), together with new family members from *Dictyostelium discoideum* (Dd) and *Arabidopsis thaliana* (At). See also ***Figure 1—figure supplements 1–4***, ***Figure 1—source data 1, 2***.

The following source data and figure supplements are available for figure 1:

**Source data 1**. Large subunit homologues found by reverse HHpred in different organisms.

**Source data 2**. Medium and small subunit homologues found by reverse HHpred in different organisms.

**Figure supplement 1**. PDB entries used to search for adaptor-related proteins.

*Figure 1. Continued on next page*

*Figure 1. Continued*

**Figure supplement 2**. Summary table of all subunits identified using reverse HHpred.

**Figure supplement 3**. Subunits that failed to be identified using reverse HHpred, but were identified by homology searching using NCBI BLAST.

**Figure supplement 4**. TSET orthologues in different species.

**Figure supplement 5**. Identification of ENTH/ANTH domain proteins and the AP complexes with which they associate, using reverse HHpred.

## TSET: a new trafficking complex

To determine whether the four new candidate subunits identified in our searches actually form a complex, we transformed *D. discoideum* with a GFP-tagged version of its small (σ-like) subunit (*Figure 2A*), and then used anti-GFP to immunoprecipitate the construct and any associated proteins from cell extracts (*Figure 2B*). Precipitates were analysed by mass spectrometry, yielding ten proteins considered to be specifically immunoprecipitated (*Figure 2—figure supplement 1a*). Two of these were the small subunit itself and its GFP tag. Three others were the remaining candidate subunits: XP_639969.1 (the β-like subunit), XP_640471.1 (the γαδεζ-like subunit), and XP_629998.1 (the μ-like subunit), confirming their presence in a complex. Quantification by iBAQ indicated that these three proteins were present in the immunoprecipitate at approximately equimolar levels (*Figure 2C*, *Figure 2—figure supplement 1A*), while the small subunit and GFP tag were in ~15-fold molar excess, probably due to overexpression.

Interestingly, two of the other proteins in the immunoprecipitate, also approximately equimolar to the three coprecipitating subunits, were XP_642289.1 and XP_637150.1. Both proteins are predicted to consist of two N-terminal β-propeller domains followed by an α-solenoid (*Figure 2D*, *Figure 2—figure supplement 1C*). This type of architecture is found in several coat components, including clathrin heavy chain, SPG11 (associated with AP-5), the α-COP and β'-COP subunits of the COPI coat (B-COPI), and the Sec31 subunit of the COPII coat (*Devos et al., 2004*). HHpred analyses show that the closest matches for both XP_642289.1 and XP_637150.1 are β'-COP, followed by α-COP. Probable orthologues of XP_642289.1 and XP_637150.1 can be found in other organisms that have the four core subunits (*Figure 1—figure supplement 4*). Because proteins with this architecture often act as a coat for transport vesicles, we hypothesize that these proteins may provide a scaffold for the newly identified heterotetramer.

The other three proteins in the immunoprecipitate, secG and vacuolins A and B, appear to be less widespread taxonomically (*Figure 2—figure supplement 2 and 3*), but are nonetheless suggestive of function. SecG is related to the plasma membrane- and endosome-associated ARNO/cytohesin family of Arf GEFs in animal cells (*Shina et al., 2010*), and also appears to be equimolar with the core complex. Vacuolins are members of the SPFH (stomatin-prohibitin-flotillin-HflC/K) superfamily. They have been shown to associate with the late vacuole just before exocytosis and also with the plasma membrane (*Rauchenberger et al., 1997*; *Gotthardt et al., 2002*), and to contribute to vacuole function (*Jenne et al., 1998*). However, the amounts of coprecipitating vacuolins were more variable, suggesting that they are less tightly associated with the complex (*Figure 2—figure supplement 1A*). Thus, like TPLATE, both SecG and the vacuolins have been implicated in membrane traffic, acting at the plasma membrane and/or endosomal compartments.

As the β-like subunit is already named TPLATE, we propose similar nomenclature for the other three subunits of the heterotetramer, relating to their relative sizes: TSAUCER, TCUP, and TSPOON. For the two associated β-propeller/α-solenoid proteins, we propose TTRAY1 and TTRAY2, and for the conserved heterohexamer, we propose the name TSET.

## Characterisation of the TSET complex in *Dictyostelium*

One of the key properties of coat proteins is their ability to cycle on and off membranes. Although by widefield fluorescence microscopy TSPOON-GFP looked diffuse and cytosolic (*Figure 2—figure supplement 1B*), TIRF imaging showed a punctate pattern, especially in the cells with lower expression,

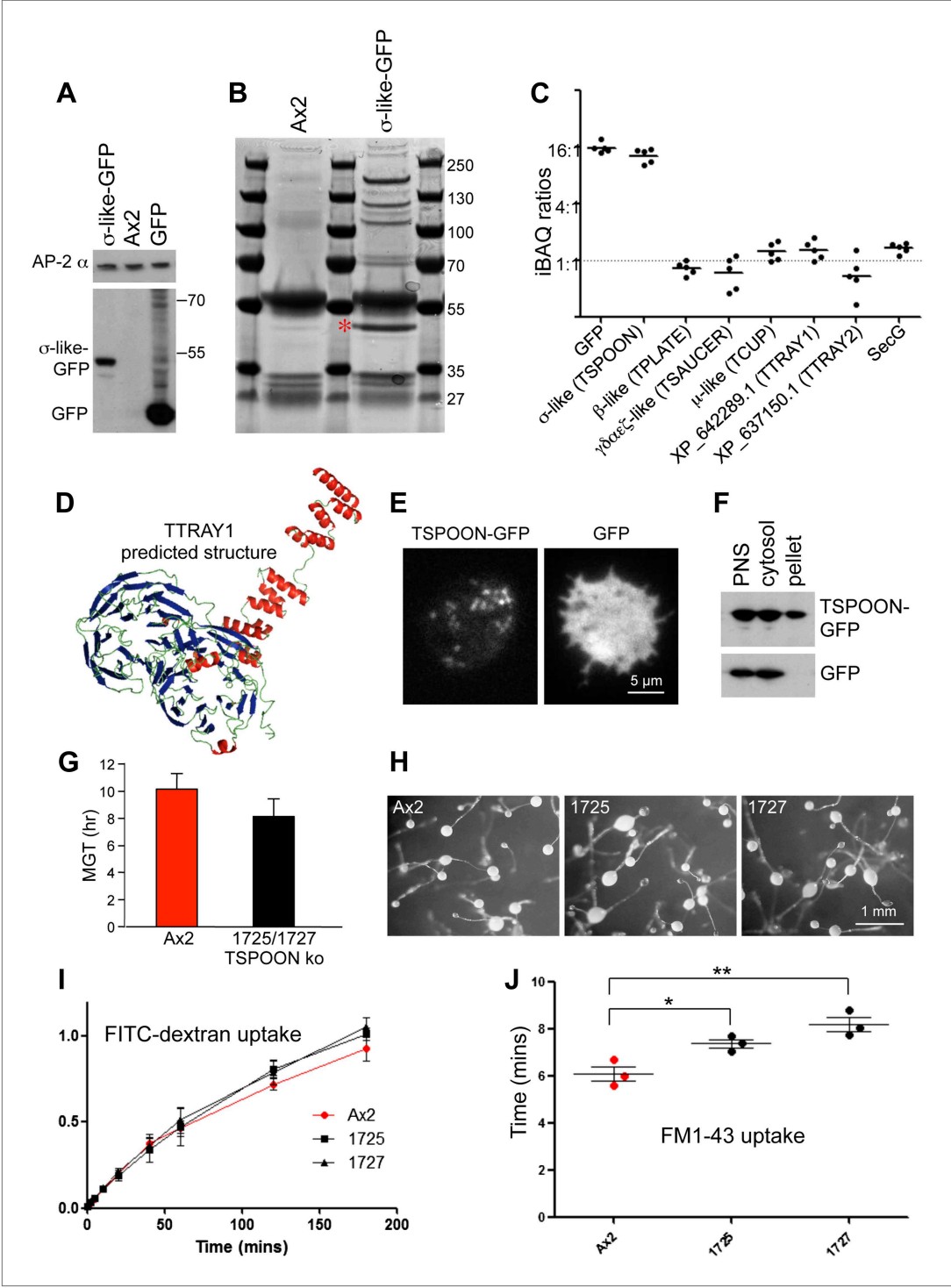

**Figure 2**. Characterisation of the TSET complex in *Dictyostelium*. (**A**) Western blots of axenic *D. discoideum* expressing either GFP-tagged small subunit (σ-like) or free GFP, under the control of the Actin15 promoter, labelled with anti-GFP. The Ax2 parental cell strain was included as a control, and an antibody against the AP-2α subunit was used to demonstrate that equivalent amounts of protein were loaded. (**B**) Coomassie blue-stained gel of GFP-tagged small subunit and associated proteins immunoprecipitated with anti-GFP. The GFP-tagged protein is indicated with a red asterix. (**C**) iBAQ ratios (an estimate of molar ratios) for the proteins that consistently coprecipitated with the GFP-tagged small subunit. All appear to be equimolar with each other, and the higher ratios for the small (σ-like/ TSPOON) subunit and GFP are likely to be a consequence of their overexpression, which we also saw in a repeat

*Figure 2. Continued on next page*

*Figure 2. Continued*

experiment in which we used the small subunit's own promoter (*Figure 2—figure supplement 1*). (**D**) Predicted structure of the N-terminal portion of *D. discoideum* TTRAY1, shown as a ribbon diagram. (**E**) Stills from live cell imaging of cells expressing either TSPOON-GFP or free GFP, using TIRF microscopy. The punctate labelling in the TSPOON-GFP-expressing cells indicates that some of the construct is associated with the plasma membrane. See *Videos 1 and 2*. (**F**) Western blots of extracts from cells expressing either TSPOON-GFP or free GFP. The post-nuclear supernatants (PNS) were centrifuged at high speed to generate supernatant (cytosol) and pellet fractions. Equal protein loadings were probed with anti-GFP. Whereas the GFP was exclusively cytosolic, a substantial proportion of TSPOON-GFP fractionated into the membrane-containing pellet. (**G**) Mean generation time (MGT) for control (Ax2) and TSPOON knockout cells. The knockout cells grew slightly faster than the control. (**H**) Differentiation of the Ax2 control strain and two TSPOON knockout strains (1725 and 1727). All three strains produced fruiting bodies upon starvation. (**I**) Assay for fluid phase endocytosis. The control and knockout strains took up FITC-dextran at similar rates. (**J**) Assay for endocytosis of membrane, labelled with FM1-43, showing the time taken to internalise the entire surface area. The knockout strains took significantly longer than the control (*p<0.05; **p<0.01). See also *Figure 2—figure supplements 1 and 2*, *Figure 2*; *Videos 1 and 2*.

The following figure supplements are available for figure 2:

**Figure supplement 1**. Further characterisation of *Dictyostelium* TSET.

**Figure supplement 2**. Distribution of secG.

**Figure supplement 3**. Distribution of vacuolins.

---

indicating that some of the construct is associated with the plasma membrane (*Figure 2E*, *Figure 2*; *Video 1*). In contrast, free GFP appeared to be entirely cytosolic (*Figure 2E*, *Figure 2*; *Video 2*). In addition, high speed centrifugation of a post-nuclear supernatant showed a substantial amount of TSPOON-GFP coming down in the membrane-containing pellet, in contrast to free GFP, which was exclusively in the supernatant (*Figure 2F*). These findings indicate that like other coat proteins, the complex is transiently recruited onto a membrane (specifically, the plasma membrane) from a cytosolic pool.

Silencing TPLATE in *Arabidopsis* produces a very severe phenotype, with impaired growth and differentiation, thought to be caused by defects in clathrin-mediated endocytosis (*Van Damme et al., 2006*; *Van Damme et al., 2011*). To investigate the function of TSET in *Dictyostelium*, we disrupted the TSPOON gene by replacing most of the coding sequence with a selectable marker (*Figure 2—figure supplement 1D*). Surprisingly, the resulting knockout cells grew at least as fast a control axenic strain (*Figure 2G* shows the mean generation time); and differentiation also appeared normal, with fruiting bodies forming under appropriate stimuli (*Figure 2H*). Uptake of FITC-dextran, an assay for fluid phase endocytosis, was unimpaired in the TSPOON knockout cells (*Figure 2I*); however, uptake of FM1-43, a membrane marker, was slower than in the control (*Figure 2J* shows the time taken to internalise the entire surface area), indicating that TSET plays a role in plasma membrane turnover, consistent with studies on *Arabidopsis*. Nevertheless, it is clear that in contrast to *Arabidopsis*, *Dictyostelium* can thrive without a functional TSET complex.

Very recently, the discoverers of TPLATE used tandem affinity purification to identify TPLATE binding partners, and found the *Arabidopsis* orthologues of the TSET components that we

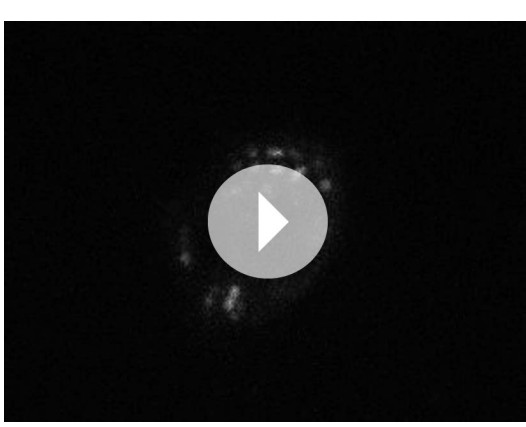

**Video 1**. Related to *Figure 2*. TIRF microscopy of *D. discoideum* expressing TSPOON-GFP, expressed off its own promoter in TSPOON knockout cells. One frame was collected every second. Dynamic puncta can be seen, indicating that the construct forms patches at the plasma membrane.

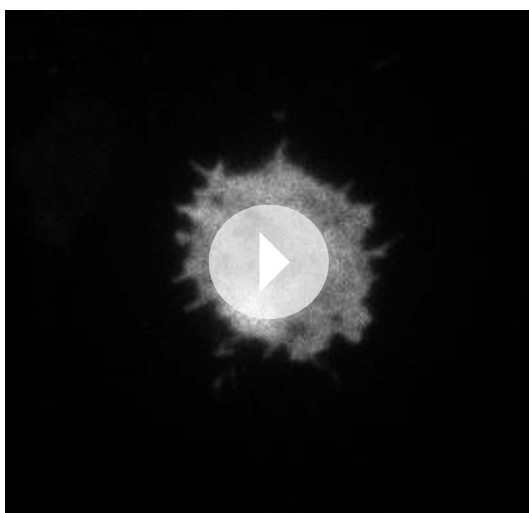

**Video 2**. Related to *Figure 2*. TIRF microscopy of *D. discoideum* expressing free GFP, driven by the Actin15 promoter in TSPOON knockout cells. One frame was collected every second. The signal is diffuse and cytosolic.

identified independently in the present study (*Gadeyne et al., 2014*). The *Arabidopsis* pull-downs did not contain any proteins resembling secG or the vacuolins, supporting our hypothesis that these proteins are add-ons to the core heterohexamer. However, *Arabidopsis* TSET is associated with two additional proteins containing EH domains, which we did not find in our *Dictyostelium* pulldowns. Some of the *Arabidopsis* pulldowns also brought down components of the machinery for clathrin-mediated endocytosis, including clathrin itself. Although we also found clathrin and associated proteins in our *Dictyostelium* immunoprecipitates, these proteins were equally abundant in control immunoprecipitates from non-GFP-expressing cells, indicating that they were contaminants. The differences in proteins that coprecipitate with TSET in the two organisms are probably a reflection of functional differences: TSET knockouts in *Arabidopsis* are lethal and knockdowns profoundly affect clathrin-mediated endocytosis, while TSET knockdowns in *Dictyostelium* produce a very mild phenotype.

## TSET is ancient and widespread in eukaryotes

When TPLATE was discovered in *Arabidopsis*, it was reported to be unique to plant species (*Van Damme et al., 2006*; *Van Damme et al., 2011*). Similarly, in the more recent *Arabidopsis* study, the authors concluded that the complex was plant-specific (*Gadeyne et al., 2014*). However, these conclusions were based on analyses of plants, yeast, and humans only. Our identification and characterization of homologues of all six subunits in *Dictyostelium discoideum*, as well as their presence in the excavate *Naegleria gruberi*, suggested that the evolutionary distribution was much more extensive. In depth homology searching identified orthologues in genomes from across the broad diversity of eukaryotes (*Figure 3*, *Figure 3—source data 1*, *Figure 3—figure supplement 1*), strongly suggesting that the complex was present prior to the LECA.

Although TSET is clearly ancient, its relationship to the other heterotetrameric complexes was unclear from homology searching alone. Consequently, after analyses of the individual subunits (*Figure 4—figure supplements 1–7*), we performed a phylogenetic analysis on the concatenated set of the four core subunits for direct comparison of TSET with the other AP and COPI complexes (*Figure 4A*, *Figure 4—figure supplement 8*). This provided moderate support for TSET as a clade, but strong resolution excluding it from the APs and COPI, as well as backbone resolution between the heterotetramer clades. Thus, TSET is clearly an ancient component of the eukaryotic membrane-trafficking system, distinct from the known heterotetramers.

Phylogenetic analysis of the TTRAYs and their closest relatives, β′-COP and α-COP (*Figure 4—figure supplement 9*), showed that the paralogues are due to ancient duplications in the TSET and COPI families respectively, which occurred prior to the divergence of the LECA. Together, these findings imply that the ancestor of the TSET, COPI, and AP complexes was a heterohexamer rather than a heterotetramer, consisting of five different proteins, with the two scaffolding proteins present as two identical copies (*Figure 4B,C*). These scaffolding subunits then duplicated independently in COPI and TSET. The ancestral AP complex may have lost its original scaffolding subunits, although AP-5, the first AP to branch away, is closely associated with SPG11, a β-propeller + α-solenoid protein whose relationship to the TTRAYs and B-COPI is as yet unclear. None of the other APs has any closely associated proteins with this architecture, but AP-1 and AP-2 transiently interact with clathrin, and there may also be a transient association between AP-3 and another β-propeller + α-solenoid protein, Vps41 (*Rehling et al., 1999*; *Cabrera et al., 2010*; *Asensio et al., 2013*).

Although TSET is deduced to have been present in LECA, the complex appears to have been entirely or partially lost in various lineages (*Figure 3B*). None of the subunits has a full orthologue in

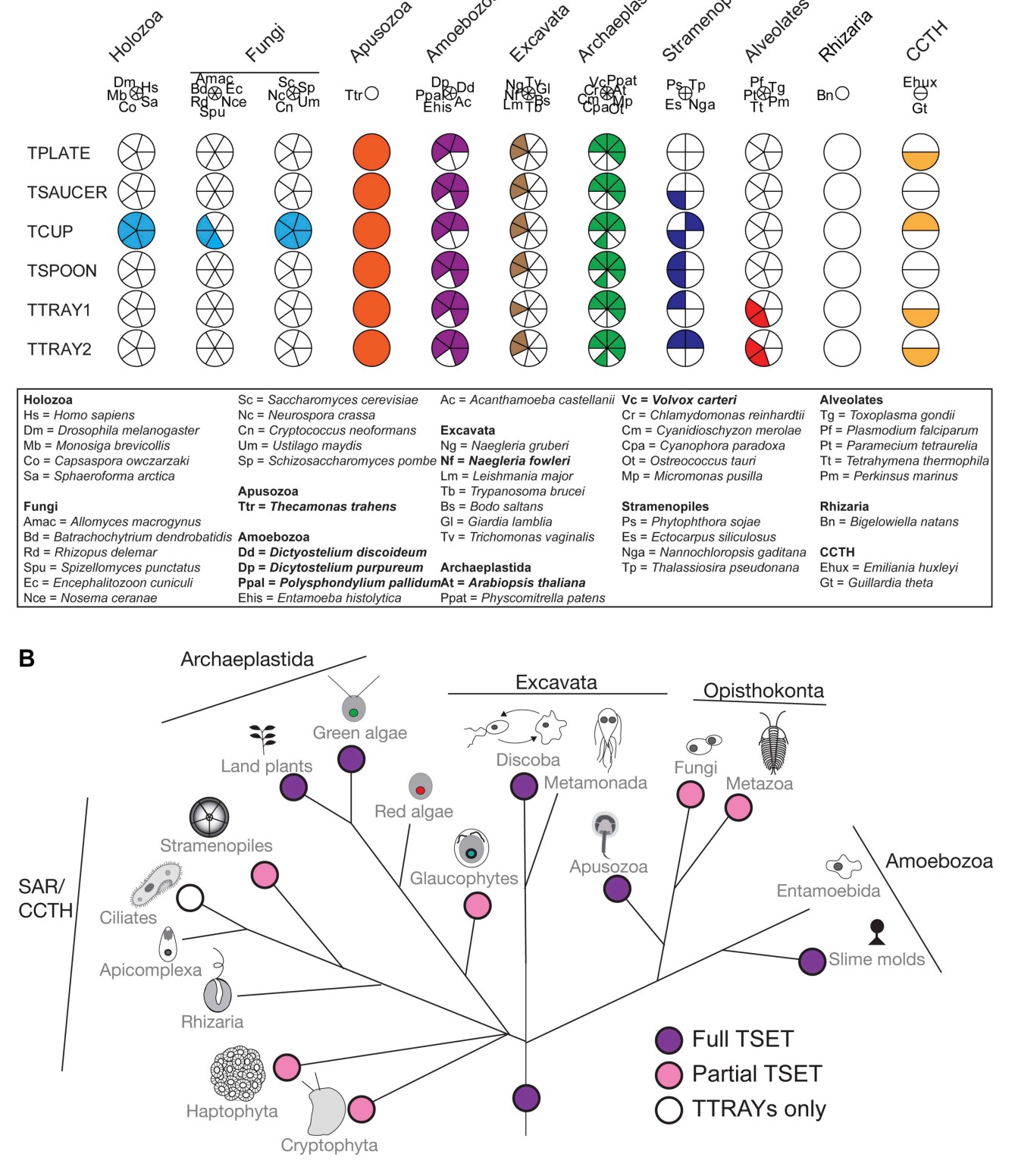

**Figure 3**. Distribution of TSET subunits. (**A**) Coulson plot showing the distribution of TSET in a diverse set of representative eukaryotes. Presence of the entire complex in at least four supergroups suggests its presence in the last eukaryotic common ancestor (LECA) with frequent secondary loss. Solid sectors indicate sequences identified and classified using BLAST and HMMer. Empty sectors indicate taxa in which no significant orthologues were

*Figure 3. Continued*

identified. Filled sectors in the Holozoa and Fungi represent F-BAR domain-containing FCHo and Syp1, respectively. Taxon name abbreviations are inset. Names in bold indicate taxa with all six components. (**B**) Deduced evolutionary history of TSET as present in the LECA but independently lost multiple times, either partially or completely. See also *Figure 3—source data 1*, *Figure 3—figure supplement 1*.

The following source data and figure supplements are available for figure 3:

**Source data 1**. Sequences used for phylogenetic analyses.

**Figure supplement 1**. Models used for phylogenetic analyses.

opisthokonts (animals and fungi), indicating secondary loss in the line leading to humans. However, the C-terminal domain of TCUP is homologous to the C-terminal domains of the muniscins, opisthokont-specific proteins (*Gadeyne et al., 2014*) (*Figure 4—figure supplement 10*). This suggests that in opisthokonts, the TCUP gene retained its 3′ end, which then combined with a new 5′ end encoding an F-BAR domain to generate the muniscin family (*Figure 4C*). These include the vertebrate proteins FCHo1/2 and the yeast protein Syp1, important players in the endocytic pathway (*Reider et al., 2009*; *Henne et al., 2010*; *Cocucci et al., 2012*; *Umasankar et al., 2012*; *Mayers et al., 2013*). The muniscins constitute one of eight families of MHD proteins in humans, and the only family whose evolutionary origin was unexplained until now. The present study indicates not only that the muniscins are homologous to TCUP, but also that they are the sole surviving remnants of the full TSET complex that existed in our pre-opisthokont ancestors.

## Conclusions

TSET is the latest addition to a growing set of trafficking proteins that have ancient distributions, but are frequently lost (*Schlacht et al., 2014*), or in the case of TSET reduced perhaps with neofunctionalization (*Figure 3*). This is consistent with the uneven distribution of the individual components (in contrast to the all-or-nothing distribution of AP-5), the additional apparently lineage-specific binding partners in *Dictyostelium*, and the acquisition of extra domains (e.g., F-BAR in opisthokonts and SH3 in plants) adding lineage-specific function.

Studies on the muniscins may help to explain the different phenotypes of TSET knockouts in *Dictyostelium* and *Arabidopsis*. Like *Arabidopsis* TSET, the muniscins interact with EH domain-containing proteins and participate in clathrin-mediated endocytosis (*Reider et al., 2009*; *Henne et al., 2010*; *Cocucci et al., 2012*; *Umasankar et al., 2012*; *Mayers et al., 2013*). *Dictyostelium* has lost its TCUP MHD, and it seems likely that concomitant with this loss, it also lost some of TSET's binding partners and functions. Nevertheless, we suspect that TSET may predate clathrin-mediated endocytosis, for two reasons. First, AP-1 and AP-2, the two AP complexes that function together with clathrin, are the most recent additions to the AP family (*Figure 4A*); and second, TSET already has its own β-propeller + α-solenoid scaffold, so it is not clear why it would need clathrin as well. Thus, the interaction between TSET and the clathrin pathway may have evolved considerably later than TSET itself, although still pre-LECA. It is tempting to speculate that TSET was part of the original endocytic machinery, which then became redundant in some organisms as the clathrin pathway took over.

Thus, our bioinformatics tool, reverse HHpred, is able to find novel homologues of known proteins, and could potentially be used to identify new players both in membrane traffic and in other pathways (*Figure 1—figure supplement 5*). Using this tool, we were able to find the four core subunits of an ancient complex belonging to the same family as the APs and COPI. This ancient complex, TSET, is therefore both the answer to the question of the origin of the last set of MHD proteins in humans, and a major new piece of the puzzle to be incorporated alongside the other membrane-trafficking machinery, as we delve into the history of the eukaryotic cell.

## Materials and methods

### Construction of the 'reverse HHpred' database

The proteomes of various organisms (detailed in *Figure 1—figure supplement 2*) were downloaded from the National Center for Biotechnology Information archives at ftp://ftp.ncbi.nih.gov/refseq/release/.

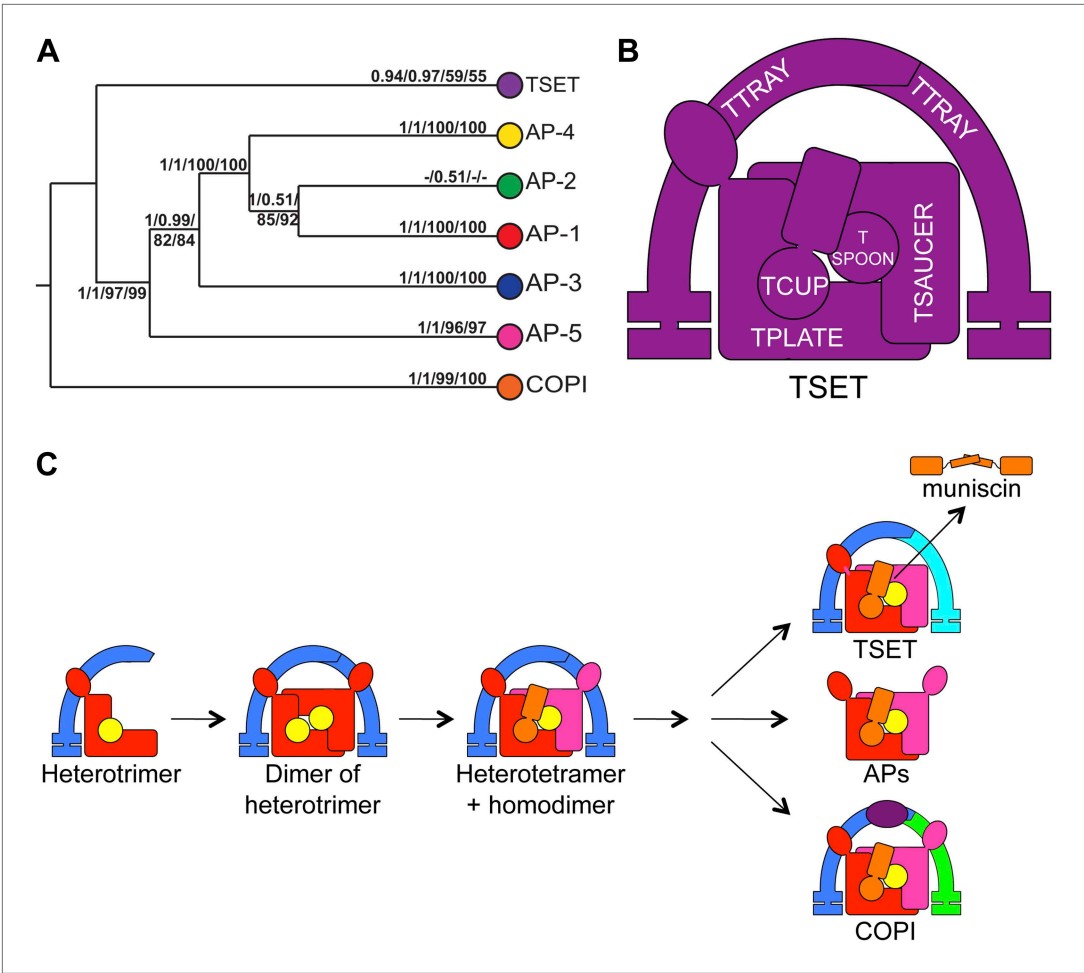

**Figure 4**. Evolution of TSET. (**A**) Simplified diagram of the concatenated tree for TSET, APs, and COPI, based on *Figure 4—figure supplement 8*. Numbers indicate posterior probabilities for MrBayes and PhyloBayes and maxium-likelihood bootstrap values for PhyML and RAxML, in that order. (**B**) Schematic diagram of TSET. (**C**) Possible evolution of the three families of heterotetramers: TSET, APs, and COPI. We propose that the earliest ancestral complex was a likely a heterotrimer or a heterohexamer formed from two identical heterotrimers, containing large (red), small (yellow), and scaffolding (blue) subunits. All three of these proteins were composed of known ancient building blocks of the membrane-trafficking system (***Vedovato et al., 2009***): α-solenoid domains in both the large and scaffolding subunits; two β-propellers in the scaffolding subunit; and a longin domain forming the small subunit. The gene encoding the large subunit then duplicated and mutated to generate the two distinct types of large subunits (red and magenta), and the gene encoding the small subunit also duplicated and mutated (yellow and orange), with one of the two proteins (orange) acquiring a μ homology domain (MHD) to form the ancestral heterotetramer, as proposed by Boehm and Bonifacino (12). However, the scaffolding subunit remained a homodimer. Upon diversification into three separate families, the scaffolding subunit duplicated independently in TSET and COPI, giving rise to TTRAY1 and TTRAY2 in TSET, and to α- and β'-COP in COPI. COPI also acquired a new subunit, ε-COP (purple). The scaffolding subunit may have been lost in the ancestral AP complex, as indicated in the diagram; however, AP-5 is tightly associated with two other proteins, SPG11 and SPG15, and the relationship of SPG11 and SPG15 to TTRAY/B-COPI remains unresolved, so it is possible that SPG11 and SPG15 are highly divergent descendants of the original scaffolding subunits. The other AP complexes are free heterotetramers when in the cytosol, but membrane-associated AP-1 and AP-2 interact with another scaffold, clathrin; and AP-3 has also been proposed to interact transiently with a protein with similar architecture, Vps41 (***Rehling et al., 1999***; ***Cabrera et al., 2010***; ***Asensio et al., 2013***). So far no scaffold has been proposed for AP-4. Although the order of emergence of TSET and COP relative to adaptins is unresolved, our most recent analyses indicate that, contrary to previous reports (***Hirst et al., 2011***), AP-5 diverged basally within the adaptin clade, followed by AP-3, AP-4, and APs 1 and 2, all prior to the LECA. This still suggests a primordial bridging of the secretory and phagocytic systems prior to emergence of a *trans*-Golgi
*Figure 4. Continued on next page*

*Figure 4. Continued*

network. The muniscins arose much later, in ancestral opisthokonts, from a translocation of the TSET MHD-encoding sequence to a position immediately downstream from an F-BAR domain-encoding sequence. Another translocation occurred in plants, where an SH3 domain-coding sequence was inserted at the 3' end of the TSAUCER-coding sequence. See also *Figure 4—figure supplements 1–10*.

The following figure supplements are available for figure 4:

**Figure supplement 1**. Phylogenetic analysis of TPLATE, β-COP, and β-adaptin, with TPLATE robustly excluded from the β-COP clade.

**Figure supplement 2**. Phylogenetic analysis of TPLATE and β-adaptin subunits (β-COP removed) showing, with weak support, that TPLATE is excluded from the adaptin clade.

**Figure supplement 3**. Phylogenetic analysis of TSAUCER, γ-COP, and γαδεζ-adaptin subunits, with TCUP robustly excluded from the γ-COP clade, and weakly excluded from the adaptin clade.

**Figure supplement 4**. Phylogenetic analysis of TSAUCER and γαδεζ-adaptin subunits (γ-COP removed), showing weak support for the exclusion of TSAUCER from the adaptin clade.

**Figure supplement 5**. Phylogenetic analysis of TCUP, δ-COP, and μ-adaptin subunits, with TSAUCER robustly excluded from the δ-COP clade and weakly excluded from the adaptin clade.

**Figure supplement 6**. Phylogenetic analysis of TCUP and μ-adaptin subunits (δ-COP removed), showing weak support for the exclusion of TCUP from the adaptin clade.

**Figure supplement 7**. Phylogenetic analysis of TSPOON with ζ-COP and σ–adaptin subunits with moderate support for the exclusion of TSPOON from both the COPI and adaptin clades, in addition to moderate support for the monophyly of the TSPOON clade.

**Figure supplement 8**. TSET is a phylogenetically distinct lineage from F-COPI and the AP complexes.

**Figure supplement 9**. Phylogenetic analysis of TTRAY1, TTRAY2, α-COP, and β'-COP.

**Figure supplement 10**. Muniscin family members identified by reverse HHpred, using the following PDB structures.

The *.protein.faa.gz files obtained were then split into separate files, each containing one protein sequence. These were stored such that each directory contained information from only one species (the total number of protein 'faa' files searched for each organism were: *Arabidopsis thaliana*, 35270; *Caenorhabditis elegans*, 23903; *Dictyostelium discoideum*, 13262; *Dictyostelium purpureum*, 12399; *Drosophila melanogaster*, 22256; *Giardia lamblia*, 6502; *Homo sapiens*, 32977; *Micromonas pusilla*, 10269; *Mus musculus*, 29897; *Naegleria gruberi* , 15756; *Physcomitrella patens*, 35893; *Saccharomyces cerevisiae*, 5882; *Schizosaccharomyces pombe*, 5004; *Selaginella moellendorffii*, 31312; *Vitis vinifera*, 23492; *Volvox carteri*, 14429). The latest protein data bank (pdb70), which contains all publicly available 3D structures of proteins, was downloaded from the Gene Center Munich, Ludwig-Maximilians-Universität (LMU) Munich via their web site at: ftp://toolkit.lmb.uni-muenchen.de/pub/HHsearch/databases/hhsearch_dbs/. The linux rpm version 2.0.11 of the hhsuite software was downloaded from the same website at ftp://toolkit.lmb.uni-muenchen.de/pub/HH-suite/releases/. Each of the faa files was then compared to the pdb70 databank using the hhsearch program from the above suite. The files were tested using the default parameters. Once each protein sequence was tested, the output file was parsed and the hits were extracted and then inserted into a mysql database. The database is searchable by keywords in PDB entries, and therefore is limited to searches where the structure of a given domain structure has been solved. The database is accessible using the link http://reversehhpred.cimr.cam.ac.uk, and searches can be initiated using keywords. Should the link become unavailable, or if you are interested in hosting this yourself please email jpn25@cam.ac.uk for more

information. A conceptually similar database, 'BackPhyre', has independently been generated, using Phyre (*Kelley and Sternberg, 2009*) rather than HHpred as a starting point to identify homologues of known proteins based on predicted structural similarities. Like reverse HHpred, BackPhyre is able to find three of the four TSET subunits in *Arabidopsis*; however, the only eukaryotes represented in BackPhyre are *A. thaliana*, *D. melanogaster*, *H. sapiens*, *M. musculus*, *P. falciparum*, and *S. cerevisiae*; and without additional organisms, such as *D. discoideum* and *N. gruberi*, we would not have been able to find the entire TSET complex.

## Data assimilation

The large adaptor subunits share sequence and structure homology, as do the medium and small subunits. Therefore, we were able to combine searches for novel large subunits, or for medium/small subunits. Using the key words 'clathrin', 'adaptor', 'adapter', 'adaptin', 'AP1', 'AP2', 'AP3', 'AP4', we searched in PDB for solved structures of any large or medium/small subunit in a given organism (11 solved structures for the large subunits and six solved structures for the medium/small subunits were used to initiate searches [*Figure 1—figure supplement 1*]). These structures span different domains found within the subunits. For each search, a list was output of any proteins found to contain structural homology. Included in this information are the precise amino acids encompassing the region of similarity, the probability score, and most importantly the 'result number'. A protein with a 'result number' of '1' means that there was no other structure in the PDB database that it is more like. Since multiple structures for the various subunits were used, we could also factor in the number of times a particular protein was identified in a search ('repeats'). These parameters were used as key pieces of evidence to determine how likely a hit in these searches would be. Once the primary data were outputted, all other manipulations were performed in Excel. For the large subunits there were 11 data sets (the 11 structures used to search for homologues), and for the medium/small subunits there were six data sets. The data manipulation was standardised at this point, and the following steps performed to assimilate the data. The data sets were sorted by result number to preclude anything with a result number of >50 (this means that there are 49 other structures in the PDB database that this protein is more similar to). Duplicates, where a protein was identified in multiple searches, were removed with the highest ranking (in 'result' terms) kept, and the number of times it was identified recorded in a new column ('repeats'). The results were the ordered with the lowest 'Result number' and the highest 'Probability' to give a final list of proteins (*Figure 1—source data 1, 2*). Generally only proteins with a 'Result number' <10, 'Probability' >50%, at least 100 amino acids of homology ('thstt' to 'thend'), and 'Repeats' at least two times were considered to be real hits. For ease of visualisation, only proteins with Result number <10 or 'Repeats' >2 are shown, and other proteins of interest (e.g., FCHo1, Syp1) with Result number <10 that did not fit the criteria listed above are greyed out. The 'IDs' have been deduced using NCBI BLAST searches, and have not been experimentally verified. Where the identity is ambiguous (such as the identity of a β-adaptin), a shared homology is suggested.

## *Dictyostelium*: the search for TSPOON and TCUP

While searching for genes encoding potential components of the complex in four dictyostelid genomes, we could find complete sets in *Polysphondylium pallidum* and *Dictyostelium fasciculatum*, but one component each was missing in the databases of predicted proteins of *D. discoideum* (σ-like subunit) and *D. purpureum* (μ-like subunit). We identified these genes by tblastn (*Camacho et al., 2009*), using the most closely related orthologous sequence as query and the chromosomal sequences as target. Gene models were created and refined using the Artemis tool (*Carver et al., 2012*). These two genes have been given the DictyBase IDs DDB_G0350235 (*D. discoideum* TSPOON) and DPU0040472 (*D. purpureum* TCUP) (www.dictybase.org).

## *Dictyostelium* expression constructs

The σ-like (TSPOON) coding sequence (CDS) was synthesised (GeneCust) with a BglII restriction site inserted at its 5′ end, its stop codon removed, and a SpeI site inserted at its 3' end, then cloned into pBluescript KSII and sequenced. The CDS was then transferred into a derivative of pDM1005 (*Veltman et al., 2009*) as a BglII/SpeI fragment, placing GFP at the C terminus, with expression driven from the constitutive actin15 promoter, to generate plasmid pJH101. In addition, the TSPOON promoter and the first 105 bases of the CDS were amplified from Ax2 gemonic DNA by PCR, using primers (5′TATCTCGAGCGTCTTCATCTTCACTATCATTTAATG-3′) and

(5'-TAAAAGCTTTTCATATTCACTCTGTTTCTCGTC-3'). The product was cut with XhoI/HindIII, and the 536-bp fragment cloned into the pBluescript KSII plasmid already containing the TSPOON CDS, via the XhoI site in the vector and the silent HindIII site introduced at nucleotide +97 of the TSPOON CDS during its synthesis. The resulting promoter-driven TSPOON CDS was removed by digestion with XhoI/SpeI and inserted into the corresponding sites of pDM323 and pDM450, resulting in expression constructs containing the TSPOON CDS with GFP fused at its C terminus and driven by its own promoter (pDT61 and pDT58 respectively).

### *Dictyostelium* cell culture and transformation

All of the methods used for cell biological studies on Dictyostelium are described in detail at Bio-protocol (*Hirst et al., 2015*). *D. discoideum* Ax2-derived strains were grown and maintained in HL5 medium (Formedium) containing 200 µg/ml dihydrostreptomycin on tissue culture treated plastic dishes, or shaken at 180 rpm, at 22°C (*Kay, 1987*). Cells were transformed with expression constructs (30 µg/$4 \times 10^6$ cells) by electroporation using previously described methods (*Knecht and Pang, 1995*). Transformants were selected and maintained in axenic medium supplemented with 60 µg/ml hygromycin (pDT58 and pJH101) and 20 µg/ml G418 (pDT61 and Actin15_GTP; *Traynor and Kay, 2007*). For the TSPOON knockout, 17.5 µg of the blasticidin disruption cassette, freed from pDT70 by disgestion with ApaI and SacII, was added to $4 \times 10^6$ Ax2 cells before electroporation. Transformants were selected and maintained in HL5 medium containing 10 µg/ml blasicidin.

### *Dictyostelium* microscopy and fractionation

Cells were transformed with GFP driven by the actin 15 promoter (A15_GFP; *Traynor and Kay, 2007*), or with TSPOON-GFP driven by either the actin 15 promoter (A15_TSPOON -GFP) or its own promoter (promoter_TSPOON-GFP). For microscopy, the cells were washed in KK2 (16.5 mM KH2PO4, 3.8 mM K2HPO4, 2 mM MgSO4) at $2 \times 10^7$/ml and then transferred into glass bottom dishes (MatTek, Ashland, MA) at $1 \times 10^6$/cm². They were either imaged immediately (vegetative) or allowed to starve for a further 6–8 hr (developed) before imaging live on a Zeiss Axiovert 200 inverted microscope (Carl Zeiss, Jena, Germany) using a Zeiss Plan Achromat 63 × oil immersion objective (numerical aperture 1.4), an OCRA-ER2 camera (Hamamatsu, Hamamatsu, Japan), and Improvision Openlab software (PerkinElmer, Waltham, MA). Various treatments including with or without starvation, fixation, pre-fixation saponin treatment did not reveal obvious membrane-associated labelling in cells expressing either promoter_TSPOON-GFP and A15_TSPOON expressing cells.

For TIRF microscopy, TSPOON-GFP was expressed in the TSPOON null cell lines HM1725 and HM1727 (see below), using the promoter_TSPOON-GFP plasmids pDT58 or pDT61. Transformants were selected and maintained in 30 µg/ml hygromycin (pDT58) or 10 µg/ml G418 (pDT61). As a control, free GFP was expressed in the null cells using the plasmid A15_GFP. Cells were harvested from tissue culture dishes when they formed a semi-confluent monoloyer and washed in KK2C (KK2 containing 0.1 mM CaCl₂). Approximately $3 \times 10^4$ cells were added to 35-mm glass bottom (No.1.5 coverglass) microwell dishes (MatTek) containing 2.5 ml of KK2C. They were incubated at 22°C for 2 hr to allow residual fluorescence associated with ingested axenic medium to dissipate, and 20 min before imaging, the KK2C was with fresh KK2C, containing 50 µg/ml L-ascorbic acid as an antioxidant to reduce the effects of phototoxicity. Cells were visualised using a Nikon N-STORM microscope operating in the TIRF mode with a 100x lens (NA 1.49) and a zoom of 1.5x.

For fractionation, cells expressing A15_GFP or promoter_TSPOON-GFP were grown until they reached a density of $2–4 \times 10^6$/ml in selective media, and by microscopy >50% of cells were expressing GFP. Starting with a maximum of $8 \times 10^8$ cells, the cells were washed in KK2 buffer and then pelleted at 600 × g for 3 min. The cells were resuspended in PBS with a protease inhibitor cocktail (Roche), lysed by 8 strokes of a motorized Potter–Elvehjem homogenizer followed by 5 strokes through a 21-g needle, and centrifuged at 4100 × g for 32 min to get rid of nuclei and unbroken cells. The postnuclear supernatant was then centrifuged at 50,000 rpm (135,700 × g RCFmax) for 30 min in a TLA-110 rotor (Beckman Coulter) to recover the membrane pellet. The cytosolic supernatant and pellet were run on pre-cast NUPAGE 4–12% BisTris Gels (Novex) at equal protein loadings, and Western blots were probed with an antibody against GFP (*Seaman et al., 2009*).

### *Dictyostelium* pulldowns and proteomics

Pulldowns were performed using *Dictyostelium discoideum* stably expressing TSPOON-GFP under a constitutive (A15_ TSPOON-GFP) and its own promoter (prom_TSPOON-GFP). Similar results were

found with both cell lines regardless of the promoter. Non-transformed cells were used as a control. Cells were grown until they reached a density of 2–4 × 10⁶/ml in selective media, and by microscopy >50% of cells were expressing GFP. Starting with a maximum of 8 × 10⁸ cells, they were pelleted by centrifugation (600×*g* for 2 min) and washed twice in KK2 buffer before being resuspended at 2 × 10⁷ cells/ml in KK2 buffer and starved for 4–6 hr at 22°C by shaking at 180 rpm. The cells were then pelleted at 600×*g* for 3 min and then lysed in 4 ml PBS 1% TX100 plus protease inhibitor cocktail tablet (Roche) for 10 min on ice, and then spun 20,000×*g* 15 min to get rid of debris and insoluble material. By protein assay the resulting lysate contained 10–15 mg total protein. The lysates were pre-cleared using PA-sepharose 30 min, and then immunoprecipitated using anti-GFP overnight with rotation at 4°C. PA-sepharose was added for 60 min and then the antibody complexes washed with PBS 1%TX100 followed by PBS before elution from beads with 100 mM Tris, 2% SDS 60°C for 10 min. The eluted proteins were precipitated with acetone overnight at −20°C, recovered by spinning 15,000×*g* 5 min and then resuspending in sample buffer. The samples were run on pre-cast NUPAGE 4–12% BisTris Gels (Novex), stained with SimplyBlue Safe Stain (Invitrogen) and then cut into 8 gel slices. Each gel slice was processed by filter-aided sample preparation solution digest, and the sample was analyzed by liquid chromatography–tandem mass spectrometry in an Orbitrap mass spectrometer (Thermo Scientific; Waltham, MA) (*Antrobus and Borner, 2011*).

Proteins that came down in the non-transformed control were eliminated, as were any proteins with less than 5 identified peptides, proteins that did not consistently coimmunoprecipitate in three independent experiments, or proteins of very low abundance compared with the bait (i.e., molar ratios of <0.002). The remaining ten proteins were considered to be specifically immunoprecipitated. Normalized peptide intensities were used to estimate the relative abundance of the specific interactors (iBAQ method; *Schwanhäusser et al., 2011*). For each protein, the values from all five repeats were plotted, including the bait protein and GFP which are clearly overrepresented by overexpression. The relative abundances of proteins were normalized to the median abundance of all proteins across each experiment (i.e., median set to 1.0) and values were then log-transformed and plotted.

## *Dictyostelium* gene disruption

The TSPOON disruption plasmid was constructed by inserting regions amplified by PCR from upstream and downstream of the TSPOON gene into both side of the blasticidin-resistance cassette in pLPBLP (*Faix et al., 2004*). The primer pair used to amplify the 5′ region was TCP1 (5′-ACTGGGCCCTGATGTTTACCTCTCTTTGGGTCATCCCATTCTATAC-3′) with σ-TCP2 (5′-AAAAAGCTTTATTACCATTGTTATTGGTAATTAACAAACTATTGATC-3′) and for the 3′ homology TCP3 (5′-A CCGCGGCCGCATAATTCAAAGAGGTCATTTAGATCAAGTTCAATTAG-3′) with TCP4 (5′-CCTCCGCGGCTTCAGGCATTGGTTCAACTTCTTGATTATTCTCAAC -3'). The PCR products were inserted as ApaI/HindIII and NotI/SacII fragments into the corresponding sites in pLPBLP, yielding pDT70.

Growth of control vs mutant strains was assayed in HL5 medium, by calculating the mean generation time, and on *Klebsiella aerogenes* bacterial lawns, by monitoring the expansion of a spot of 10⁴ cells. Spore viability was also assayed, both with and without detergent treatment, by clonally diluting spores on bacterial lawns and counting the resultant plaques (*Kay, 1982*).

## Endocytosis assays

Membrane uptake was measured in real time at 22°C with 2 × 10⁶ cells in 1 ml of KK2C containing 10 μM FM1-43 (Life Technologies). Briefly, a 2-ml fluorimeter cuvette containing 0.9 ml of KK2C plus 11 μM FM1-43 was placed in the fluorimeter (PerkinElmer LS50B) with stirring set on high. The uptake was initiated by the addition of 100 μl cells at 2 × 10⁷/ml in KK2C and data collected every 1.2 s at an excitation of 470 nm (slit width 5 nm) and emission of 570 nm (slit width 10 nm) for up to 360 s. The uptake curves were biphasic and the data were normalized against the initial rise in fluorescence, when the cells were first added to the FM1-43, as this essentially corresponds to the dye incorporation into the plasma membrane only (*Aguado-Velasco and Bretscher, 1999*). The uptake rate was calculated from linear regression of the initial linear phase of the uptake using GraphPad Prism software. The surface area uptake time is 1/slope of the initial phase.

Fluid phase uptake was measured at 22°C using FITC-dextran 70 kDa (Sigma FD-70) by adding 2 mg/ml (final) to cells (1 × 10⁷/ml) in filtered HL5 medium that was shaken at 180 rpm. Duplicate

0.5 ml samples were taken at each time point and diluted in 1 ml of ice-cold HL5 in a microcentrifuge tube held on iced water. Cells were pelleted, the supernatant aspirated, and the pellet washed twice by centrifugation in 1.5 ml ice-cold wash buffer (KK2C plus 0.5%wt/vol BSA) before being lysed in 1 ml of buffer [100 mM Tris–HCl, 0.2% (vol/vol) Triton X-100, pH 8.6] and fluorescence then determined (excitation 490 nm, slit width 2.5 nm; emission 520 nm, slit width 10 nm). Data were normalized to protein content (*Traynor and Kay, 2007*).

## Comparative genomics

Sequences from *Arabidopsis thaliana*, *Dictyostelium discoideum*, and *Naegleria gruberi* were obtained with our new reverse HHpred tool. These sequences were used to build HMMs for each subunit using HMMer v3.1b1 (http://hmmer.org). HMMs were used to search the protein databases for the organisms in *Figure 3A* (see *Figure 3—source data 1* for the location of each genomic database). Sequences identified as potential homologues were verified through reciprocal BLAST into the genomes of each of the original three sequences. Sequences were considered homologues if they retrieved the correct orthologue as the reciprocal best hit in at least one of the reference genomes, with an e-value at least two orders of magnitude better than the next best hit. New sequences were incorporated into the HMM prior to searching a new genome in order to increase the sensitivity and specificity of the HMM. Genomic protein databases were also searched by BLAST using the closest related organism with an identified sequence as the reference genome. Nucleotide databases (scaffolds or contigs) were also searched using tblastn to ensure that no sequences were missed resulting from incomplete protein databases. The distribution of TSET components is displayed in Coulson plot format using the Coulson plot generator v1.5 (*Field et al., 2013*).

## Phylogenetic analysis

Identified sequences were combined with the adaptin and COPI sequences from *Hirst et al. (2011)* into subunit-specific data sets with the intention of concatenation. Data sets were aligned using MUSCLE v3.6 (*Edgar, 2004*) and masked and trimmed using Mesquite v2.75. Phylogenetic analysis was carried out using MrBayes v.3.2.2 (*Ronquist and Huelsenbeck, 2003*) and RAxML v7.6.3 (*Stamatakis, 2006*), hosted on the CIPRES web portal (*Miller et al., 2010*). MrBayes was run using a mixed model with the gamma parameter until convergence (splits frequencey of 0.1). RAxML was run under the LG + F + CAT model (*Lartillot et al., 2009*) and bootstrapped with 100 pseudoreplicates. The resulting trees were visualized using FigTree v1.4. Initial data sets were run and long branches were removed. Data sets were then re-aligned and re-run as above. Opisthokont adaptin and COPI sequences were also removed from all data sets except from the TCUP alignment. Data sets were realigned and new phylogenetic analyses were carried out. Remaining sequences were used for concatenation. Sequences were aligned and trimmed, as above, and concatenated using Geneious v7.0.6. Subsequent phylogenetic analysis was carried using PhyloBayes v3.3 (*Lartillot et al., 2009*) under the LG + CAT model until a splits frequency of 0.1 and 100 sampling points was achieved, and PhyML v3.0, with model testing carried out using ProtTest v3.3. MrBayes and RAxML were used as above. Raw phylogenetic trees were converted into figures using Adobe Illustrator CS4. The models of amino acid sequence evolution are provided in *Figure 3—figure supplement 1*. The database identifiers of all sequences and their abbreviations and figure annotations are provided in *Figure 3—source data 1*. All alignments are available in *Supplementary file 1*.

## Homology modeling

The Phyre v2.0 web server (*Kelley and Sternberg, 2009*) was used to predict the 3D structures of each TTRAY from *A. thaliana*, *D. discoideum,* and *N. gruberi*. Default settings were used for structural predictions, and structures were visualized using MacPyMOL (www.pymol.org).

## Acknowledgements

We thank the members of the Robinson, Dacks, and Kay labs for helpful discussions.

JH and JPN conceived the database, JH collated all the searches and performed some of the functional analysis in *Dictyostelium*, JBD, and AS performed the in depth evolutionary analysis, DT and RRK conceived and performed the majority of the experiments in *Dictyostelium*, GB helped to identify *Dictyostelium* orthologues, and RA performed the mass spectrometry analysis. MSR, JBD and JH prepared the manuscript.

## Additional information

### Funding

| Funder | Grant reference number | Author |
|---|---|---|
| The Wellcome Trust | 086598 | Jennifer Hirst, Robin Antrobus, Margaret S Robinson |
| Medical Research Council | MC_U105115237 | David Traynor, Gareth Bloomfield, Robert R Kay |
| Alberta Innovates Technology Futures | New Faculty Award 201000076 | Alexander Schlacht, Joel B Dacks |
| Natural Sciences and Engineering Research Council of Canada | PGSD3-442728-2013 | Alexander Schlacht |

The funders had no role in study design, data collection and interpretation, or the decision to submit the work for publication.

### Author contributions

JH, AS, JBD, MSR, Conception and design, Acquisition of data, Analysis and interpretation of data, Drafting or revising the article; JPN, DT, GB, RA, RRK, Conception and design, Acquisition of data, Analysis and interpretation of data

## Additional files

### Supplementary file

• Supplementary file 1. Untrimmed, masked alignments used in phylogenetic analysis (*Figure 4A*, and Supplemental Figures. Masks indicated regions of sequences retained for phylogenetic analysis. Text files containing aligned sequences are in FASTA format. Alignment for the concatenated tree is composed of the trimmed TPLATE.R2, TSAUCER.R2, TCUP.R2, and TSPOON.R2 alignments.

### Major datasets

The following dataset was generated:

| Author(s) | Year | Dataset title | Dataset ID and/or URL | Database, license, and accessibility information |
|---|---|---|---|---|
| Hirst J, Schlacht A, Norcott JP, Traynor D, Bloomfield G, Antrobus R, Kay RR, Dacks JB, and Robinson MS | 2014 | Reverse HHpred | http://reversehhpred.cimr.cam.ac.uk | Publicly available at the given URL. |

The following previously published datasets were used:

| Author(s) | Year | Dataset title | Dataset ID and/or URL | Database, license, and accessibility information |
|---|---|---|---|---|
| Pruitt KD, Brown GR, Hiatt SM, Thibaud-Nissen F, Astashyn A, Ermolaeva O, Farrell CM, Hart J, Landrum MJ, McGarvey KM, Murphy MR, O'Leary NA, Pujar S, Rajput B, Rangwala SH, Riddick LD, Shkeda A, Sun H, Tamez P, Tully RE, Wallin C, Webb D, Weber J, Wu W, Dicuccio M, Kitts P, Maglott DR, Murphy TD, Ostell JM | 2014 | RefSeq | ftp://ftp.ncbi.nih.gov/refseq/release/ | Publicly available at the given URL. |
| Bernstein FC, Koetzle TF, Williams GJ, Meyer EE Jnr, Brice MD, Rodgers JR, Kennard O, Shimanouchi T, and Tasumi M | 1977 | pdb70 | ftp://toolkit.lmb.uni-muenchen.de/pub/HHsearch/databases/hhsearch_dbs/ | Publicly available at the given URL. |

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
