## [Decision Letter]

Thank you for sending your work entitled “TSET: an ancient and widespread trafficking complex” for consideration at *eLife*. Your article has been favorably evaluated by a Senior editor and 2 reviewers, one of whom is a member of our Board of Reviewing Editors.

The Reviewing editor and the other reviewer discussed their comments before we reached this decision, and the Reviewing editor has assembled the following comments to help you prepare a revised submission.

This manuscript describes the characterization of new heterohexameric complex (TSET) from the slime mold *Dictyostelium*, with striking similarity to the AP and F-COPI adaptor complexes. Only using a clever 'reverse HHpred' strategy did the authors find the complex, since its sequence conservation is too weak for identification using more classical methods (Blast or the like). After biochemically characterizing the complex the authors go on with an in-depth phylogenetic analysis, which makes the clearest case yet that all APs, F-COPI, and TSET likely diverged from a common ancestor already present in LECA. This is a significant advance in our understanding of the evolution of eukaryotic life in general, and vesicle-mediated trafficking within the endomembrane system in particular. Importantly, this study also corrects the highly visible recently published misconception of a plant-specific adaptor complex (Gadeyne et al., Cell 2014).

Neither of the reviewers had major concerns and both strongly recommended publication. The conclusions are based on rigorous experimental procedures and skilled phylogenetic and bioinformatic analysis. However, one of the reviewers raised two minor points that you should consider in the revised manuscript:

1) The authors describe the 'reverse HHpred' method as novel; however, it appears to be conceptually very similar to the BackPhyre procedure, except that other proteomes are searched (http://www.sbg.bio.ic.ac.uk/phyre2/html/help.cgi?id=help/backphyre). The authors are encouraged to reword the relevant text passages about the 'novelty' of reverse HHpred accordingly or, better, compare and contrast with BackPhyre.

2) The protocoatomer hypothesis as originally stated in [10] posited that the NPC and vesicle coats are related because of the use of proteins with common architecture in all systems. Ten years later, with an enormous amount of new data, we know that the story is not quite so simple. The beta-propeller is one of the most common folds in eukaryotes, and alpha-solenoids are also found in many processes, often to generate large protein assemblies. It is still an open question to what extend these different coating systems have a common origin, since the assembly of the only very superficially similar 'protocoatomer' structures are drastically different in clathrin COPI, COPII, and the NPC. To support the main thrust of this paper, we think that invoking the overly simplistic protocoatomer hypothesis is rather misleading, thus we would suggest to take it out or only refer to it in the Discussion.

---

## [Author Response]

*1) The authors describe the 'reverse HHpred' method as novel; however, it appears to be conceptually very similar to the BackPhyre procedure, except that other proteomes are searched (*http://www.sbg.bio.ic.ac.uk/phyre2/html/help.cgi?id=help/backphyre*). The authors are encouraged to reword the relevant text passages about the 'novelty' of reverse HHpred accordingly or, better, compare and contrast with BackPhyre*.

One of the reviewers pointed out that our reverse HHpred tool is conceptually similar to BackPhyre. We hadn’t known about BackPhyre, so thank you for bringing it to our attention. We have now taken out the word “new” when referring to our tool, both in the Introduction and in the Conclusions; and in the Materials and methods section, we now compare and contrast reverse HHpred and BackPhyre. Interestingly, like reverse HHpred, BackPhyre was able to find three of the four core TSET subunits in *Arabidopsis*, and in both cases the subunit that couldn’t be found was TSAUCER. However, because reverse HHpred contains sequences from many more eukaryotes than BackPhyre, were able to find TSAUCER in both *Naegleria* and *Dictyostelium*, neither of which is represented in BackPhyre. So without HHpred, we would never have been able to identify the entire heterotetramer bioinformatically. Because the BackPhyre website encourages users to submit requests for additional genomes to be added, eventually it may be possible to use reverse HHpred and BackPhyre as complementary tools for searching the entire diversity of eukaryotes.

*2) The protocoatomer hypothesis as originally stated in*
[10]
*posited that the NPC and vesicle coats are related because of the use of proteins with common architecture in all systems. Ten years later, with an enormous amount of new data, we know that the story is not quite so simple. The beta-propeller is one of the most common folds in eukaryotes, and alpha-solenoids are also found in many processes, often to generate large protein assemblies. It is still an open question to what extend these different coating systems have a common origin, since the assembly of the only very superficially similar 'protocoatomer' structures are drastically different in clathrin COPI, COPII, and the NPC. To support the main thrust of this paper, we think that invoking the overly simplistic protocoatomer hypothesis is rather misleading, thus we would suggest to take it out or only refer to it in the Discussion*.

The same reviewer made the point that the 10-year-old protocoatomer hypothesis, which we refer to several times in our original manuscript, is an oversimplification, because lots of proteins that are not vesicle coats also have the β-propeller and α-solenoid architecture. Thus, we have removed all mention of protocoatomers per se, while still pointing out that this type of architecture is commonly used when building the scaffolding part of a coat, and still citing the Devos et al. paper that first made this point. We do suspect that at least some of the coat scaffolds have a common origin. For instance, an HHpred analysis of Sec31 pulls out β’-COP immediately after Sec31 itself, although the relationship between clathrin heavy chain, Sec31, and COPI appears to be more distant. An in-depth phylogenetic analysis should shed more light on the evolution of these proteins, but for the time being, we agree that it is best to use the word “scaffold” rather than “protocoatomer”.

In addition to the changes described above, we have added some new TIRF microscopy data showing *Dictyostelium* expressing TSPOON-GFP. The original version of our manuscript contained only widefield data, which showed a diffuse cytosolic distribution of GFP-tagged TSPOON (which we know is overexpressed). However, by TIRF microscopy, we see discrete puncta, especially in the lower-expressing cells. This provides further evidence for TSPOON acting at the plasma membrane in *Dictyostelium* as well as in plants.

We have also changed the title, from “TSET: an ancient and widespread trafficking complex” to “TSET: an ancient and widespread membrane trafficking complex” because one of our colleagues pointed out that for those not in the membrane traffic field, it wouldn’t be clear from the title or abstract what the paper was going to be about.